# Sesquiterpene Lactones as Promising Anti-Glioblastoma Drug Candidates Exerting Complex Effects on Glioblastoma Cell Viability and Proneural–Mesenchymal Transition

**DOI:** 10.3390/biomedicines13010133

**Published:** 2025-01-08

**Authors:** Andrey V. Markov, Arseny D. Moralev, Kirill V. Odarenko

**Affiliations:** Institute of Chemical Biology and Fundamental Medicine, Siberian Branch of the Russian Academy of Sciences, Lavrent’ev Avenue 8, 630090 Novosibirsk, Russia; arseniimoralev@gmail.com (A.D.M.); k.odarenko@yandex.ru (K.V.O.)

**Keywords:** brain tumor, terpenes, tumor cell death, mitochondrial dysfunction, epithelial–mesenchymal transition, glial–mesenchymal transition, polypharmacology

## Abstract

Glioblastoma is one of the most aggressive brain cancers, characterized by active infiltrative growth and high resistance to radiotherapy and chemotherapy. Sesquiterpene triterpenoids (STLs) and their semi-synthetic analogs are considered as a promising source of novel anti-tumor agents due to their low systemic toxicity and multi-target pharmacological effects on key processes associated with tumor progression. The current review aims to systematize the knowledge on the anti-glioblastoma potential of STLs accumulated over the last decade and to identify key processes in glioblastoma cells that are most susceptible to the action of STLs. An analysis of published data clearly demonstrated that STLs, which can successfully cross the blood–brain barrier, exert a complex inhibitory effect on glioblastoma cells through the induction of the “mitochondrial dysfunction–oxidative stress–apoptosis” axis, the inhibition of glucose metabolism and cell cycle phase transition, and the suppression of glioblastoma cell motility and invasion through the blockade of proneural–mesenchymal transition. Taken together, this review highlights the promising anti-glioblastoma potential of STLs, which are not only able to induce glioblastoma cell death, but also effectively affect their diffusive spread, and suggests the possible directions for further investigation of STLs in the context of glioblastoma to better understand their mechanism of action.

## 1. Introduction

Glioblastoma multiforme (GBM) is one of the most aggressive forms of brain cancer, with an incidence ranging from 1 to 155 per 1,000,000 individuals in people younger than 20 years and older than 75 years, respectively [1]. Despite the fact that patients with glioblastoma receive a full range of therapies, including deep tumor resection, tumor bed radiotherapy, and chemotherapy, the vast majority of these patients do not survive beyond 18 months [2], and their median progression-free survival (PFS) is only about 7 months from the time of diagnosis [3]. This alarming fact is directly related to the active diffuse spread of glioblastoma cells into healthy brain tissue, which makes it virtually impossible to completely remove tumor cells by maximal safe resection [4], and to the rapid acquisition of resistance to radiotherapy and chemotherapy by glioblastoma cells [5], which mediates a high probability of glioblastoma recurrence. Separately, it is necessary to note the extremely limited list of chemopreparations approved for glioblastoma therapy in the clinic, which is limited to alkylating agents, including temozolomide, and nitrosoureas, such as carmustine, lomustine, and fotemustine, and the anti-angiogenic humanized monoclonal antibody bevacizumab, the efficacy of which remains low, in part due to the high intertumoral heterogeneity of glioblastoma [6] and the ability of the tumor microenvironment to stimulate proneural–mesenchymal transition (PMT; also known as glial–mesenchymal transition), a process by which glioblastoma cells acquire a highly invasive mesenchymal phenotype that is resistant to therapeutic intervention [7].

Given the complex nature of glioblastoma pathogenesis with an extensive network of master regulators [8], in addition to the development of selective inhibitors of glioblastoma-related signaling proteins, the search for multi-target anti-glioblastoma drugs is of great interest and includes two main directions, namely, drug repurposing (see recent comprehensive reviews [9,10]) and the synthesis of drug candidates based on multi-target natural metabolites [11]. In the latter case, among the wide variety of structural scaffolds used for the development of nature-based anti-glioblastoma agents, sesquiterpene lactones (STLs) deserve special attention due to their pronounced complex action against both tumor cells [12] and the tumor microenvironment [13,14,15], their ability to modulate a wide range of signaling pathways involved in glioblastoma pathogenesis, including PMT [12,16], and their ability to efficiently accumulate in brain tissues [17]. Despite the experimental data confirming the anti-glioblastoma potential of STLs, including their inhibitory effects on key glioblastoma-related processes, including glioblastoma cell motility and invasion [18,19,20], clonogenicity [21], expression of PMT marker proteins [22,23], and expansion of glioblastoma stem cells [24], there are still no comprehensive reviews summarizing, structuring, and analyzing this information. Only Yan et al. recently described the anti-glioma mechanisms of these compounds [19], but this review remains inaccessible to the broad scientific community because it is published only in Chinese. Some recent reviews provide brief information on the anti-glioblastoma activities of STLs, mainly describing their cell death-inducing effects [20,25,26], which do not allow us to assess the prospects of this class of compounds as blockers of glioblastoma cell infiltration.

In this review, we summarize the recent findings on the complex inhibitory effects of STLs, both natural and semi-synthetic, on glioblastoma cells published in the last decade and answer the question of which processes in these cells are most susceptible to STLs. In addition, the pharmacological potential of STLs in relation to PMT was separately discussed.

## 2. Anti-Glioblastoma Activity of Sesquiterpene Lactones

### 2.1. Anti-Tumor Potency of STLs

Sesquiterpene lactones (STLs) are a broad class of secondary natural metabolites, mainly found in plants of the *Asteraceae* family, formed from three isoprene linkages and containing an α,β-unsturated α-methylene-γ-lactone ring, which allows STLs to react by Michael-type addition with biological nucleophiles (e.g., sulfhydryl groups of glutathione and cysteine residues), thereby determining their multi-target action on key processes associated with various pathologies, including malignant growth [27,28]. Based on the structure of the carbon skeleton, STLs can be classified into seven major types, including germacranolides with a ten-membered ring, three types with a six-membered ring, namely, elemanolides, eudesmanolides, and eremophilanolides, and guaianolides, pseudoguaianolides, and hypocretenolides with five- and seven-membered rings [14]. These compounds have promising complex anti-tumor effects against a wide range of tumors of different histogenesis, not only by causing tumor cell death through the induction of apoptosis [25,26,29], necroptosis [30], ferroptosis [31], cytodestructive autophagy [32], and endoplasmic reticulum stress [33], but also by effectively blocking their capacity for high motility and invasion [34], spheroid growth [35], colony formation [36], adhesion to extracellular matrix [37], and vasculogenic mimicry [38], as well as by disrupting tumor cell energetics [39]. The pronounced anti-tumor potency of STLs has been demonstrated not only in vitro, but also in animal models, including murine xenograft models of non-small lung cancer [40], breast cancer [41], hepatocellular carcinoma [42], glioblastoma [18], neuroblastoma [33], and others. Currently, a number of STLs and their derivatives are undergoing clinical trials, including mipsagargin in patients with advanced solid tumors (Phase 1, completed) [43] and artesunate in patients with cervical intraepithelial neoplasia (Phase 1, completed) [44]. Thus, the accumulated material demonstrates the high anti-tumor potential of STLs and their ability to modulate various processes associated with tumor growth and metastasis.

### 2.2. Criteria for Selection of Published Material and Structure of Analyzed Data

To perform a detailed analysis of the anti-glioblastoma activity of STLs, we used Google scholar and PubMed search engines to search for experimental articles published from 2014 to 2024 using the queries “sesquiterpene lactone” and “glioblastoma” or “glioma”, followed by the selection of papers containing information on the direct pharmacological action of STLs on glioblastoma cells in vitro and in vivo. A total of 47 publications were identified that met the search criteria, describing the anti-glioblastoma activity of 28 natural and 5 semi-synthetic STLs belonging to germacranolides (12 molecules), eudesmanes (2 molecules), guaianolides (12 molecules), edoperoxide-containing STLs (2 molecules), and STLs with other structural types (5 molecules) (Figure 1). A careful analysis of these reports revealed that the main attention of researchers has been focused on the analysis of STL-induced cell death, primarily their apoptogenic activity, while the effects of STLs on the processes associated with the infiltrative growth of glioblastoma, including PMT, have been much less studied (Table 1). Because STLs generally have similar effects on the same characteristics of glioblastoma cells regardless of their structural affiliation (Table 1), in this review, we have systematized the accumulated knowledge by focusing on STL-sensitive processes.

### 2.3. Direct Toxic Effect of PTLs on Glioblastoma Cells: Key Processes and Mechanisms

STLs were shown to be effective in suppressing glioblastoma cell viability in vitro, with cytotoxic effects mainly below 50 μM at 24–72 h incubation with a median IC_50_ (50% inhibitory concentration) of 16 μM (Table 1). Notably, STLs inhibited the viability not only of various glioblastoma cells but also of patient-derived glioblastoma stem cells (GSCs) [24,45], demonstrating the high pharmacological potential of STLs against heterogeneous glioblastoma populations.

**Table 1 biomedicines-13-00133-t001:** Anti-glioblastoma activity of STLs.

Type	Compound	Cell line	Concentration	Biological Effects	Effects on Cell Signaling	Ref.
Germacranolides	Parthenolide (PTL)	U373	8–16 μM	Cytotoxicity (IC_50_^(24 h)^~17 μM); ↓ ^1^ survivin, G_2_/M arrest, ↑ ^2^ phosphatydilserine (PS) externalization, ↑cleaved caspase-3, ↑cytoplasm vacuoles, ↑LC3-II/LC3-I	↓Cdk2, ↑Chk2, ↑ULK1	[46]
	Dimethylaminoparthenolide (DMAPT)	9LSF	5–25 μM 40 mg/kg, i.p. ^3^	In vitro: cytotoxicity (IC_50_^(48 h)^~7 μM) In vivo (BBB permeability): effective uptake by orthotopic 9LSF tumor in rats	ND ^4^	[47]
		U87, GBM6, GL261	1–10 μM; 100 mg/kg (30 times)	In vitro: cytotoxicity (IC_50_ = 3.5–8.8 μM) In vivo (BBB permeability): accumulation in brains of healthy mice (6251 ng/g (1 h); brain-to-plasma ratio: 2.1 (1 h) and 3.0 (4 h)) In vivo (GL261; orthotopic): ↑survival	ND	[48]
	Tanacin (TC)	U87	1–20 μg/mL	Cytotoxicity (IC_50_^(48 h)^ = 4.5 μg/mL)	ND	[49]
	Isocostunolide (ICTL)	Glioma stem cell lines: GSC-3#, GSC-12#, GSC-18#	0.1–10 μg/mL	Cytotoxicity (IC_50_^(72 h)^ = 1.1–2.8 μg/mL), ↑PS externalization, ↑cleaved caspase-3, ↓spheroidal growth, ↓colony formation capacity	ND	[45]
	Elephantopinolide A (EPA)	U87	1–50 μM	Cytotoxicity (IC_50_^(48 h)^ = 4.22 ± 0.11 μM), ● ^5^ GSTP1 (molecular docking (MolDock), thermal shift assay), ↓GSTP1, ↑PS externalization, ↑chromatin condensation, apoptosis (acridine orange/ethidium bromide staining), ↑cleaved caspase-7, ↑Bax, ↓Bcl-xl, ↓Δψ_M_ ^6^, oxidative stress: ↑mitochondrial ROS, ↑H_2_O_2_, ↑OH, ↑lipid peroxidation	↓JNK1 (mRNA, protein), ↑p-JNK, ↑p-STAT3	[50]
	Costunolide (CTL)	A173, U87	10–40 μM 5 mg/kg, i.p. (10 times)	In vitro: ROS-dependent cytotoxicity (IC_50_^(24 h)^~30 μM), ↑ROS, ↓telomerase activity, ↓hTERT, ↑p53, ↑caspase-3/8 activity, ↑Bax/Bcl-2, ↓glucose metabolism: ↓G6PD, ↓TKT, ↓TKT activity; ↑senescence: ↑β-gal-cell staining, ↑GS(P), ↑glycogen accumulation In vivo (U87, heterotopic): ↓tumor volume, ↓tumor weight, ↓telomerase activity, ↑ROS, ↑caspase-3/8 activity, ↓TERT, ↓G6PD, ↓TKT, ↓GS(P)	↓Nrf2	[51]
	Molephantin (MPT)	U251, U87	3–100 μM 10 and 30 mg/kg, i.p. (10 times)	In vitro: cytotoxicity (IC_50_^(72 h)^ = 10.6–22.6 μM), ↓colony-forming capacity, S arrest, ↓migration, ↓invasion, ↓vimentin, ↓Snail, ↓N-cadherin, ↑E-cadherin, ↑PS externalization, ↑Bax/Bcl-2, ↑ROS, ↑cleaved caspase-7/9/3, ↑cleaved PARP, ↑mitochondrial ROS, ↓Δψ_M_, ↑mitochondrial dynamic imbalance (↓Mfn1/2, ↓OPA1, ↑Fis1, ↑Drp1, ↑mitochondrial fragmentation), ↓late stage mitophagy, ↓autophagosome–lysosome fusion, ↓spheroid growth In vivo (U87, heterotopic): brain tissue accumulation, ↓tumor volume, ↓tumor weight, no organ toxicity, ↑Bax/Bcl-2, ↑cleaved caspase-9/7/3, ↑cleaved PARP	↓CDK4, ↓CDK2, ↑p21, ↓p-PI3K, ↓p-Akt, ↓p-mTOR,	[52]
	Melampomagnolide B dimer (MPLBD)	9L-SF	3–10 μM	Cytotoxicity	ND	[53]
	1	C6	ND	IC_50_ = 3.0 ± 0.8 μM	ND	[54]
	2	U87, MC38	1–16 μM 40 mg/kg, p.o. ^7^ (12 times)	In vitro: cytotoxicity (IC_50_^(96 h)^ = 2.8 μM), ↑PS externalization, G_0_/G_1_ arrest In vivo (MC38, heterotopic): ↓tumor weight, no organ toxicity	●NF-κB (MolDock)	[55]
	Parthenolide dimer (смoтри 5) (PTLD)	U87, U118, SF126, SHG44, U251, C6	1–10 μM 50 mg/kg, i.p. (6 times)	In vitro: cytotoxicity (IC_50_^(72 h)^ = 1.66–7.93 μM), ↓clonogenicity, ↑PS externalization, ↓migration, ↓invasion, ●PKM2 (molecular docking, thermal shift assay), ↑E-cadherin, ↓vimentin, ↑Bax/Bcl-2, ↓Bcl-xl In vivo (U118, heterotopic): ↓tumor volume, ↓tumor weight, ↑Bax/Bcl-2, ↑E-cadherin, ↓vimentin, ↓STAT3 MDCK cells: ↓barrier integrity	↓STAT3, ↓p-STAT3, ↑PDK4	[56]
	DMAPT-D6	U87, LN229	2.5–40 μM	ROS-dependent cytotoxicity (IC_50_ = 11.15–15.5 μM), ↓clonogenicity, S arrest, ↑ROS, ↑DNA damage (↑γH2AX, ↑p53, ↑53BP1, ↑LIG IV), ↑PS externalization, ↑cleaved caspase-3, ↑cleaved PARP	↓cyclin B, ↓cyclin E, ↓CDK1, ↓CDK2, ↑p27, ↑Nrf2, ↑DR3, ↑DR5, ↑FADD, ↑TRADD	[57]
Eudesmanes	Alantolactone (ALN)	U87, U251, U118	1–50 μM 10 and 20 mg/kg, i.p. (15 times)	In vitro: cytotoxicity (IC_50_^(48 h)^ = 16.33–29.16 μM), ↓clonogenicity, G_0_/G_1_ arrest, ↓migration, ↓invasion, ↓MMP-2, ↓MMP-9, ↑PS externalization, ↑cleaved caspase-3/9, ↑cleaved PARP, ↑Bax/Bcl-2, ↑cytoplasmic cytochrome C (cytoC), ↓COX-2 In vivo (U87, heterotopic): ↓tumor weight, ↓tumor volume, ↓COX-2, ↓p-p65 In vivo (BBB penetration): present in cerebrospinal fluid	↓Cyclin D1, ↓CDK4, ↓binding of NF-κB p50/p65 and p300 to COX-2 promoter, ↓nuclear translocation of p65/p50, ↓p-IκB-α, ↓p-IKKβ, ↓IKKβ kinase activity, ●IKKβ (MolDock)	[58]
		U87, U251	10 μM 20 mg/kg, i.p (15 times)	In vitro: ↑G-actin, ↓F-actin, ↑mitochondrial transition of F-actin, ↓p-cofilin, ↑mitochondrial transition of cofilin, ↓migration, ↓invasion, ↓MMP-2, ↓MMP-9, ↑PS externalization, ↑cleaved caspase-3/9, ↑cleaved PARP, ↑cytoC In vivo (U87, heterotopic): ↓p-cofilin, ↓p-LIMK1/2	↓p-LIMK1/2	[59]
		HCM3, U87, U251	10–50 μM 20 mg/kg, i.p. (7 times)	In vitro: cytotoxicity (IC_50_ = 10–30 μM), ↓spheroid growth, ↓CD133, ↓OCT4, ↓SOX2, ↓NANOG In vivo (U87, orthotopic): ↑survival, ↓tumor size, ↓p-EGFR, ↓p-YAP	↓YAP, ↑p-YAP, ↓p-EGFR, ↑p-LATS1	[60]
	2α-Hydroxyalantolactone (HALN)	U87, U87ΔEGFR	0.1–100 μM	Cytotoxicity (IC_50_ = 15.15–49.22 μM)	ND	[61]
Guaianolides	Micheliolide (MCL)	U251	2.5–20 μM	Cytotoxicity (IC_50_^(48 h)^ = 12.5±1.6 μM), ↓filopodia formation, ↓clonogenicity, ↓migration, ↓invasion, ↓MMP-9, ↓N-cadherin, ↓vimentin, ↑PS externalization, ↑cytoC, ↑cleaved caspase-3/9, ↑Bax/Bcl-2, ↓COX-2	↓p-IκBα/ IκBα	[62]
	Dimethylaminomicheliolide (DMAMCL, ACT001)	C6, U87	1–120 μM 25–100 mg/kg, p.o. (21 times)	In vitro: cytotoxicity (IC_50_^(72 h)^ = 20.58–27.18 μM), ↑PS externalization, ↑Bax/Bcl-2 In vivo (C6, orthotopic): ↓tumor weight, ↑survival, no organ and brain toxicity In vivo (BBB permeability): effectively accumulated in brain tissue (19.0±9.6 μg/mL (0.5 h))	ND	[63]
		U118, U251, U87, SF126, SHG44	5–40 μM	Cytotoxicity (IC_50_^(48 h)^ = 17.9–37.1 μM), ↓clonogenicity, ●PKM2 (micheliolide (DMAMCL metabolite), pull-down), ↑pyruvate kinase activity, ↓aerobic glycolysis, and ↓pentose phosphate pathway (↓lactate, ↓glucose-6-phosphate, ↓sedoheptulose-7-phosphate, ↓glycerol-3-phosphate)	ND	[21]
		GL261	0.2–100 μM 50 mg/kg, p.o. + 2 Gy X-ray (5 times)	In vitro: ↑susceptibility of GL261 cells to X-ray, ↑ROS, ↑DNA damage, ↑cleaved caspase-3 In vivo (GL261, heterotopic): ↓tumor volume (+X-ray)	ND	[64]
		U87	10 μM 200 mg/kg, p.o. (6 times)	In vitro: ●PAI-1 (proteomic analysis, thermal shift assay, pull-down, surface plasmon resonance (K_D_ = 2.31 mM), MolDock), ↓migration, ↓invasion, ↓vasculogenic mimicry, ↑PS externalization, synergy with cisplatin, ↑E-cadherin, ↓vimentin, ↓Snail, ↓β-catenin In vivo (U118, heterotopic): ↓tumor size, ↓tumor weight	↓p-PI3K, ↓p-Akt	[65]
		U251, TJ905	20–80 μM 400 mg/kg/day, p.o.	In vitro: ↓PD-L1 In vivo (GL261, orthotopic): ↑survival, ↓PD-L1, ↓p-STAT3, ↓M2 macrophage infiltration	↓p-STAT3, ●STAT3 (pull-down)	[15]
		U118, U251, SF126, SHG44, GL261	0.1–100 μM 200 and 400 mg/kg/day, p.o.	In vitro: cytotoxicity (IC_50_ = 7.3–77.3 μM), G_2_/M arrest, ↑PS externalization. ↑ROS, ↑NOX1, ↑TrX, ↑HO1, ↓MnSOD In vivo (U118, heterotopic): ↓tumor weight, ↓tumor volume In vivo (GL261, orthotopic): ↓tumor volume, ↓CDC2, ↓cyclin B1, ↓p-p65, ↓MnSOD, ↓Ki67	●IKKβ (pull-down, LC-MS/MS), ↓IKKγ, ↓p- IKKβ, ↓p-IκBα, ↓p-p65, ↓p-p65 nuclear translocation, ↓β-TRCP, ↑Nrf2	[66]
		patient-derived GSC 1123, R39	1–100 μM 100 mg/kg, p.o. (5 times)	In vitro: cytotoxicity (IC_50_ = 15.87–19.88 μM), ↓spheroid growth, ↓AEBP1, ↓TGF-β-induced parameters (↓AEBP1, ↓p-Akt, ↓ cell proliferation, ↓spheroid growth), synergy with SHP099 In vivo (1123, orthotopic): ↓tumor growth, ↑survival, ↓p-Akt, ↓AEBP1, ↓Ki67, ↓Nestin	↓p-Akt	[24]
	MCL3	G442, U87, U251, Hs683	3–30 μM 10, 20, and 40 mg/kg, p.o. (14 times)	In vitro: cytotoxicity (IC_50_^(96 h)^ = 6.44–18.90 μM), ↑PS externalization, ↓IL6, ↓HIF-1α, ↓MMP-2, ↓Bcl-2, ↓Mcl-1 In vivo (G442, heterotopic): ↓tumor volume, ↓tumor weight, ↓PCNA, ↓CD34 (angiogenesis), ↓IL6	↓p-NF-κB, ↓nuclear p-NF-κB, ↓p-STAT3, ↓nuclear p-STAT3	[67]
	Dehydrocostus lactone (DCL)	U118, U251, U87	1–100 μM 10 and 20 mg/kg, i.p. (14 times)	In vitro: cytotoxicity (IC_50_^(48 h)^ = 17.16–26.42 μM), ↓clonogenicity, ↓migration, ↑cytoC, ↑Bax/Bcl-2, ↓COX-2, ↓p300/p50/p65 NF-κB nuclear translocation, ↓p300/p50/p65 NF-κB binding to COX-2 promoter In vivo (U87, heterotopic): ↓tumor volume, ↓tumor weight, ↓COX-2, ↓p-p65, ↓p-IKKβ In vivo (BBB permeability): effectively accumulated in brain	↓p-IKKα/β, ↓p-IκBα, ↓p-p65, ●IKKβ (MolDock)	[18]
	Brevilin A (BVA)	U87, U373, LN229	5–80 μM	Cytotoxicity (IC_50_^(24 h)^ = 30–40 μM), ↑PS externalization, ↑ROS, ↓GSH, ↑Bak, ↓Bcl-xl, ↑cytoC, ↓Δψ_M_, ↑cleaved caspase-3/9, ↑cleaved PARP, ↓XIAP	↓p-JNK, ↓p-p38	[68]
	Xanthatin (XTN)	C6, U251	1–15 μM 10, 20 and 40 mg/kg, i.p. (ND)	In vitro: cytotoxicity (IC_50_^(24 h)^~15 μM), ↑PS externalization, ↑TUNEL-positive cells, ↑cleaved caspase-3, ↑Bax/Bcl-2, ↑ER stress (↑GRP78, ↑XBP1s, ↑nuclear translocation of CHOP) In vivo (C6, heterotopic): ↓tumor weight, ↑necrotic areas, ↑p-IRE1, ↑ATF6, ↑p-EIF2α, ↑XBP1s, ↑ATF4, ↑CHOP, ↑cleaved caspase-3	↑p-IRE1α, ↑p-EIF2α, ↑ATF4	[69]
		C6, U251	1–15 μM 10, 20 and 40 mg/kg, i.p. (14 times)	In vitro: cytotoxicity (IC_50_^(12 h)^~15 μM), ↓PCNA, ↓clonogenicity, ↑cleaved PARP, ↑cleaved caspase-3, ↓LC3-II/LC3-I, ↓autophagosome formation, ↑p62, ↓Beclin-1, ↓*BECN1*, ↓*ATG5*, ↓*ATG7*, ↓*ATG12* In vivo (ND, ND): ↓LC3-II/LC3-I, ↑p62, ↑Beclin-1, ↑p-Akt, ↑p-mTOR, ↓p-ULK1	↑p-Akt, ↑p-mTOR, ↓p-ULK1, no effect on p-ERK1/2, p-JNK, and p-p38	[70]
	Lactucopicrin (LPN)	U87	1–10 μM	Cytotoxicity (IC_50_^(24 h)^ = 12.5±1.1 μM), ↓clonogenicity, ↓migration, autophagy induction (↓p62, ↑LC3-II, rearrangement of vimentin and α-tubulin cytoskeleton), G_2_/M arrest, ↑p53, ↓pro-caspase-6, ↑cleaved PARP, synergy with temozolomide (TMZ)	↓p-Akt, ↓p-ERK1/2, ↑p21, ↓CDK2, ↓p65 NF-κB	[22]
	Tomentosin (TTN)	U87	5–100 μM	Cytotoxicity (IC_50_^(48 h)^ = 28.8 μM), ↑*BAX*, ↑*CASP3*, ↑*CASP8*, ↑*CASP9*, ↑*CYCS*, ↑*FADD*, ↑*TNF*, ↑*TNFR1*, ↑*TNFR2,* ↑*TIMP2*, ↓clonogenicity	ND	[71]
	Cynaropicrin (CPN)	U87	4–10 μM	ROS-dependent cytotoxicity (IC_50_^(48 h)^ = 12.8±3.3), ↓clonogenicity, ↑ROS, ↓Δψ_M_, ↑cytoC, ↓pro-caspase-9/3, ↑LC3-II/I, ↓p62, ↑senescence (↑β-gal-positive cells), additive effect with TMZ	↓p-ERK, ↓p-p65 NF-κB, ↑nuclear translocation of Nrf2	[72]
	Cynaropicrin (CPN) Dehydrocstus lactone (DCL) Saussureamine B (SAB)	U251 CSCs, U251	ND	Cytotoxicity (U251 CSCs: IC_50_ = 7.9–20.4 μM; U251: IC_50_ = 4.0–10.9 μM)	ND	[73]
	Arglabin diethyl cyanomethylphosphonate (ADCMP)	T98G	ND	Cytotoxicity (IC_50_^(48 h)^ = 16.9±1.3 μM), selectivity index = 3.2	ND	[74]
	9-Oxomicheliolide (OMCL)	U87	ND	Cytotoxicity (IC_50_ = 13.15 μM), ↑PS externalization	ND	[75]
Endoperoxide- containing STLs	Dihydroartemisinin (DAMN)	GL261 GL261 GSCs	10–80 μM	Cytotoxicity (GL261: IC_50_^(24 h)^~80 μM, GL261 GSCs: IC_50_^(24 h)^~40 μM), ↓spheroid growth, G_1_ arrest, ↑cleaved caspase-3	↓p-Akt	[76]
		U87	5–160 μM	Cytotoxicity (IC_50_~70 μM), ↓migration, ↓invasion, ↓ADAM17	↓p-EGFR, ↓p-Akt	[77]
		LN-229, LN-Z308, T269	5–9 μM	Cytotoxicity, ↓clonogenicity, synergy with TMZ, ↑ROS, ↑CAT, ↑GPX1, ↑GPX4, ↑SOD2, ↑LC3-II, ↓Sox2, ↓Nestin	ND	[78]
		U87, U251	50–600 μM 2, 10, and 50 mg/kg, p.o. (45 times)	In vitro: Cytotoxicity (IC_50_^(24 h)^ = 200–210 μM), ↓migration, ↓invasion, ↓MMP9, ↓MMP9 activity, ↓MMP7, ↓MMP7 activity, ↑ROS, ↑p53, ↑p-p53 In vivo (U87, U251, heterotopic): ↓tumor volume	↓EGFR, ↓β-catenin, ↓p-β-catenin	[79]
		U87, U251	3.125–200 μM 100 mg/kg, i.p. (28 times)	In vitro: Cytotoxicity (IC_50_^(24 h)^ = 16.12–25.05 μM), ↓DNA synthesis, ↓clonogenicity, S and G_2_/M arrest, ↑PS externalization, ↑caspase-3, ↑cleaved caspase-3/9, ↑cleaved PARP, ↓Δψ_M_, ↓glucose uptake, ↓L-lactate, ↓glycolytic capacity, synergy with TMZ, ↓spheroid formation In vivo (U87, orthotopic): ↑median survival time, ↑caspase-3	↓*PGC-1α*, ●ERRα (MolDock, TR-FRET)	[80]
	Artesunate (AST)	LN229, A172	15 μM	Senolytic activity: ↓proliferation of senescent cells, ↑PS externalization in senescent cells; non-toxic for non-senescent cells	ND	[81]
Other	3	U251, C6	1–100 μM	Cytotoxicity (IC_50_ = 36.6–41.6 μM), ↑activated caspases, ↑sub-G_0_/G_1_, ↑PS externalization, ↑ROS, ↓Δψ_M_	ND	[82]
	4	U251	5–8 μM 2 mg/kg, p.o. (14 times)	In vitro: cytotoxicity (IC_50_ = 1.7 μM), mitotic catastrophe, ↓c-Myc, ↓Bcl-2, ↓Mcl-1, ↓Bcl-xl, ↓spheroid growth, ↑spheroid cell disaggregation, ↓migration from the spheroid, ↓Hsp105, ↓vimentin, ↓TNAP2, ↓G6PD, ↓GCN1, ↓TrxR1 In vivo (U251, heterotopic): ↓tumor volume, ↓p-STAT3, ↓STAT3	↓STAT3 DNA-binding activity, ↓p-STAT3, ●STAT3 (NMR, MolDock)	[83]
	Goyazensolide (GZD)	U87, T98G	10–100 μM	Cytotoxicity (IC_50_~6 μM), ↓clonogenicity, no effect on migration, ↑apoptosis, ↑cleaved caspase-3	ND	[84]
	Deoxyelephantopin (DEN)	GL261	0.5–2 μg/mL 10 mg/kg, p.o. (14 times)	In vitro: Cytotoxicity (IC_50_^(24 h)^~2 μg/mL), ↑PS externalization, G_2_/M arrest, ↓VEGF, ↓TGF-β, ↑caspase-3, ↑Bax/Bcl-2, ↑cytoC In vivo (GL261, heterotopic): ↓tumor volume, ↓tumor weight, ↑survival	↓CDK4, ↓cyclin D2, ↓p-Akt, ↓p-STAT	[85]
	Enhydrin (EHN)	U87, LN229	2–8 μM 15–25 μM, intracraneal	In vitro: cytotoxicity (IC_50_^(24 h)^ = 1.6–2.6 μM), ↓migration, ↓invasion, ↓N-cadherin, ↓vimentin, ↑E-cadherin In vivo (ND, orthotopic): ↓tumor size, ↑survival, ↓Ki67, ↓Jun, ↓TGF-β1, ↑Smad7	↓Jun, ↓TGF-β1, ↓p-Smad2, ↓nuclear p-Smad2, ↓p-Smad3, ↓nuclear p-Smad3, ↑Smad7	[86]

^1^ ↓—inhibition or decrease of parameter; ^2^ ↑—activation or increase of parameter; ^3^ i.p.—intraperitoneal; ^4^ ND—no data; ^5^ ●—direct interaction of STLs with protein targets; ^6^ ΔψM—mitochondrial membrane potential; ^7^ p.o.—per oral.

#### 2.3.1. Pro-Apoptotic Effect of STLs on Glioblastoma Cells

Most of the reports analyzed indicate the apoptotic character of cell death under the action of STLs, including germacranolides (parthenolide [46], its derivatives parthenolide dimer [56], DMAPT-D6 [57], and 2 [55], isocostunolide [45], elephantopinolide A [50], costunolide [51], and molephantin [52]), guaianolides (micheliolide [62], DMAMCL [63,64,65,66], MCL3 [67], dehydrocostus lactone [18], brevilin A [68], xanthatin [69,70], lactucopicrin [22], tomentosin [71], cynaropicrin [72], and 9-oxomicheliolide [75]), eudesmane-type alantolactone [58,59], endoperoxide-containing dihydroartemisinin [76,80], and artesunate [81], as well as goyazensolide [84], deoxyelephantopin [85], and compounds 3 [82] and 4 [83]. The effect of these compounds on glioblastoma cells was accompanied by a number of apoptosis hallmarks, including phosphatidylserine externalization [45,46,50,52,55,56,57,62,63,67,68,69,70,71,72,75,76,80,81,82,83,85], the cleavage of poly(ADP-ribose) polymerase 1 (PARP-1) [22,52,57,58,59,68,70,80], and the activation of executioner caspases, among which the activation of caspase-3 was predominantly observed [45,46,50,52,55,56,57,62,63,67,68,69,70,71,72,75,76,80,81,82,83,85]; only elephantopinolide A and molephantin were found to activate caspase-7 [50,52] and lactucopicrin-cleaved pro-caspase-6 [22]. The activation of caspase-3/-7 was preceded by the cleavage of pro-caspase-9 in most published reports (Table 1), suggesting that the mitochondrial pathway of apoptosis is triggered by STLs.

These data are consistent with the ability of STLs to modulate the expression of Bcl-2 family proteins that control mitochondrial membrane permeability in glioblastoma cells [87], including increasing the Bax/Bcl-2 ratio [18,51,52,56,58,62,63,69,85], down-regulating Mcl-1 [67,83], and up-regulating Bak [68]. These perturbations resulted in the massive loss of mitochondrial membrane potential [50,52,68,72,80,82] and the release of cytochrome C from mitochondria into the cytosol [18,58,59,62,68,72,85] with the subsequent activation of the abovementioned caspase-9–caspase-3/-7 apoptosis axis (Table 1). Interestingly, on the background of inducing mitochondrial apoptosis in glioblastoma cells, costunolide and tomentosine also effectively activated caspase-8 [51], and DMAPT-D6 significantly activated the expression of death receptors DR3 and DR5 and adapter proteins FADD and TRADD [57,71], suggesting the ability of STLs to induce glioblastoma cell death also by intrinsic receptor-dependent apoptosis, although this effect of STLs is still poorly understood. In addition to activating the caspase cascade, STLs stimulated the expression of p53, a key apoptosis-related regulator whose suppressed function promotes glioblastoma progression [88]. Pronounced p53-inducing activity in glioblastoma cells has been shown for costunolide [51], DMAPT-D6 [57], lactucopicrin [22], and dihydroartemisinin [79], with the latter also increasing the phosphorylation level of p53 [79], a necessary step for the achievement of p53-mediated transcriptional programs [89].

Despite the stimulatory effect of STLs mainly on mitochondrial apoptosis, the subtle mechanism of this action is not fully understood. According to the accumulated published data, apoptosis-associated cytochrome C release under STL treatment occurs as a result of changes in the expression profile of Bcl-2 family proteins, primarily an increase in the Bax/Bcl-2 ratio, leading to the opening of mitochondrial transition pores (MTPs) (Table 1). In addition, parthenolide and dihydroartemisinin have been found to induce Bak oligomerization, which is necessary for proper MTP formation [90], and to interact directly with cytochrome C, probably increasing its cytoplasmic translocation [91]. Notably, some reports have demonstrated a different mechanism of action of STLs in mitochondria: it has been shown that a number of STLs, including tapsigargin [92], 4-cinnamoyloxy-1,3-dihydroxyeudesm-7,8-ene [93], and 1,2,3-triazole-bearing derivatives of dehydrocostus lactone and alantolactone [94], can interact directly with isolated mitochondria, leading to MTP opening and subsequent dissipation of the mitochondrial membrane potential. Interestingly, costunolide and artemisinin did not possess such ability with respect to mammalian mitochondria [95,96], but the latter caused the massive depolarization of isolated malarial mitochondria [96]. Thus, the mitochondria-targeting mechanism of action of STLs requires further detailed investigation.

#### 2.3.2. Effects of STLs on Proliferation of Glioblastoma Stem Cells (GSCs)

STLs have been shown to have promising effects on the proliferation of glioblastoma stem cells (GSCs), primarily by targeting key molecular pathways critical for GSCs’ survival and maintenance. Mechanistically, STLs have been shown to induce apoptosis [45,76] in GSCs by modulating signaling pathways such as PI3K/AKT/mTOR [24] and MAPK/ERK [76], which are frequently deregulated in glioblastoma. In addition, STLs have been observed to inhibit the NF-κB pathway, thereby reducing the expression of NF-κB-dependent genes associated with proliferation and stemness maintenance [18,22,55,58,62,66,67,72]. Together with the proven direct effects on GSCs, STLs were shown to suppress the stemness phenotype of GBM cells via the down-regulation of key GSCs markers, such as CD133, OCT4, SOX2, and NANOG [60].

However, the available data on the inhibitory effect of STLs on GSCs are limited by several studies and require further research to achieve a more detailed understanding. The research in this direction seems to be very promising and expedient, since the enhanced self-renewal capacity of GSCs and their ability to initiate and maintain glioblastoma growth significantly attenuate the anti-glioblastoma efficacy of chemotherapy and radiation [1,97,98]. Consequently, the targeting of GSCs has the potential to improve the currently available glioblastoma treatment and improve patient outcomes [99].

#### 2.3.3. Pro-Oxidant Effect of STLs in Glioblastoma Cells

The above data suggest a pronounced mitochondria-targeting effect of STLs underlying their potent anti-glioblastoma potency. More detailed studies revealed that STLs not only disrupted the homeostasis of Bcl-2 family proteins, but also affected processes controlling mitochondrial dynamics in glioblastoma cells. Alantolactone was found to cause the dephosphorylation of cofilin with its subsequent translocation to mitochondria together with G-actin in U87 cells [59]. Given the key regulatory function of cofilin in actin assembly, which plays an important role in mitochondrial fission, mitochondrial shape changes, and mitophagy [100], this translocation induced cofilin–actin pathology leading to mitochondrial dysfunction [101], accompanied by cytochrome C release [102]. Molephantin was also found to modulate the levels of proteins that control mitochondrial dynamics, inhibiting the expression of Mfn1/2 and OPA1 and up-regulating Fis1 and Drp1, leading to mitochondrial fragmentation and massive mitochondrial depolarization in glioblastoma cells [52].

Considering the close association of mitochondrial imbalance with reactive oxygen species (ROS) hyperproduction [103], a significant part of the published research reported the ability of STLs to induce oxidative stress in glioblastoma cells (Figure 2A). A high stimulatory activity for ROS hyperproduction was found for germacranolides (elephantopinolide A [50], costunolide [51], molephantin [52], and DMAPT-D6 [57]), guaianolides (DMAMCL [64,66], brevilin A [68], and cynaropicrin [72]), dihydroartemisinin [78,79], and lactone 3 [82]. In addition, molephantin and elephantopinolide A significantly increased mitochondrial ROS production [50,52], and the latter also induced massive lipid peroxidation in U87 cells [50]. Notably, the blockade of oxidative stress by N-acetylcysteine, a known ROS scavenger, significantly reduced the pro-apoptotic effect of costunolide [51], molephantin [52], DMAPT-D6 [57], and DMAMCL [66], suggesting the critical importance of ROS hyperproduction for the anti-glioblastoma potential of STLs.

Oxidative stress induced by STLs in glioblastoma cells was accompanied by the depletion of intracellular glutathione (GSH) [68] and the activation of compensatory cytoprotective mechanisms, including the up-regulation of Nrf2, a key transcription factor regulating cellular defense against oxidative insults [104], induced by DMAPT-D6 [57] and DMAMCL [66], and the cynaropicrin-stimulated enhancement of Nrf2 nuclear translocation in U87 cells [72]. Furthermore, a number of STLs led to the overexpression of key Nrf2-dependent anti-oxidant proteins in glioblastoma cells, namely, NADPH oxidase 1 (NOX1), thioredoxin (TrX), heme oxygenase 1 (HO1), and superoxide dismutase 2 (SOD2) by DMAMCL [66] and catalase (CAT), glutathione peroxidases 1 and 4 (GPX1/4), and SOD2 by dihydroartemisinin [78]. Interestingly, costunolide conversely inhibited the expression of Nrf2 in the background of a pronounced pro-oxidant effect in glioblastoma cells [51], and lactone 4 was shown to down-regulate the expression of thioredoxin reductase 1 (TrxR1) [83], which may indicate the ability of STLs to modulate the complex system of negative feedback loops involved in the anti-oxidant response in glioblastoma cells [105]. Considering the higher susceptibility of brain to oxidative stress compared to other organs [106] and the cellular redox imbalance found in glioblastoma cells [107], the demonstrated ability of STLs to induce ROS hyperproduction due to the disruption of mitochondrial homeostasis in glioblastoma cells (Figure 2A) can be exploited for the development of novel anti-glioblastoma drug candidates.

#### 2.3.4. Effect of STLs on Energy Metabolism of Glioblastoma Cells

In addition to the detrimental effect of STLs on mitochondrial homeostasis leading to the impairment of oxidative phosphorylation [108], a number of STLs have been found to inhibit glucose metabolism [21,51,56], confirming the ability of STLs to suppress glioblastoma cell viability and by disrupting their energy system (Figure 2A). Ahmad et al. found that costunolide significantly retarded the pentose phosphate pathway in A172 and U87 glioblastoma cells and reduced the expression of its two main enzymes, glucose-6-phosphate dehydrogenase (G6PD) and transketolase (TKT) [51]. Consistent with these results, Guo et al. confirmed the ability of DMAMCL to suppress not only the pentose phosphate pathway but also anaerobic glycolysis in U118 cells, as evidenced by the significant decrease in intracellular levels of lactate, glucose-6-phosphate, and sedoheptulose-7-phosaphate after DMAMCL treatment, as well as the down-regulation of the expression of genes encoding key glycolytic enzymes [21]. The revealed inhibitory potency of DMAMCL with respect to anaerobic metabolism in U118 cells was found to be mediated by the direct interaction of micheliolide, a major metabolite of DMAMCL, with pyruvate kinase M2 (PKM2), resulting in the increase in pyruvate kinase activity and subsequent suppression of lactate production in glioblastoma cells [21]. Interestingly, parthenolide dimer 4 was also found to bind directly to PKM2 and induce its activity in U87 and U118 cells [56], which may demonstrate the PKM2-targeting potential of different STLs independent of their structure, which requires further studies. Given the involvement of PKM2 not only in metabolic regulation but also in the control of mitochondrial dynamics and mitochondrial membrane permeability [109], the direct interaction of STLs with PKM2, along with the inhibitory effect on energy metabolism, may underlie the STL-induced mitochondrial dysfunction in glioblastoma cells discussed above.

#### 2.3.5. Effect of STLs on Other Processes Associated with Glioblastoma Cell Proliferation

The propidium iodide staining of STL-treated glioblastoma cells clearly demonstrated that the anti-glioblastoma potential of STLs was not only determined by apoptosis induction but also by the suppression of cell proliferation through cell cycle arrest (Table 1). Interestingly, this effect was not limited by the influence of these compounds on a specific cell cycle phase. It was shown that STLs led to the accumulation of glioblastoma cells in three different cell cycle phases, namely, G0/G1 arrest after treatment with alantolactone [58] and lactone 2 [55], S arrest induced by molephantin [52] and DMAPT-D6 [57], and G2/M arrest induced by parthenolide [46], DMAMCL [66], lactucopicrin [22], and deoxyelephantopin [85]. Interestingly, dihydroartemisinin blocked all of these cell cycle phases [76,80], which may be explained by the cellular context. The cell cycle-inhibitory effect of STLs was determined by their modulation of the expression of key regulators of cell cycle phase transition, including the down-regulation of cyclin-dependent kinases (CDK1 [57], CDK2 [22,46,52,57,66], and CDK4 [52,58,85]), cyclins (cyclin B [57,66], cyclin E [57], cyclin D1 [58], and cyclin D2 [85]), and the checkpoint kinase Chk2 [46], as well as the up-regulation of the cyclin-dependent kinase inhibitors p21 [22,52] and p27 [57] (Figure 2A).

Consistent with the cell cycle-targeting effect of STLs (Table 1) and the known association of cell cycle arrest with cell senescence [110], a number of research groups demonstrated the ability of STLs to induce the senescence of glioblastoma cells. Ahmad et al. [51] and Rotondo et al. [72] demonstrated a significant increase in β-galactosidase activity, a classical senescence inducer [111], in glioblastoma cells treated with costunolide and cynaropicrin, respectively. Moreover, the senescence-inducing effect of costunolide was accompanied by the intracellular accumulation of glycogen and a marked decrease in telomerase activity [51]. Interestingly, artesunate, which was not tested for its effect on glioblastoma cell senescence, significantly decreased the viability of temozolomide-induced senescent LN299 and A172 cells showing a high senolytic activity [81]. Since senescence is currently considered not only as a glioblastoma-suppressing mechanism due to its arresting effect on cell proliferation [112] but also as a factor promoting glioblastoma progression [113], further detailed studies on the senescence-modulating effect of STLs in glioblastoma cells are required.

In addition to apoptosis induction and cell cycle arrest, the anti-glioblastoma activity of STLs may also be determined by their effect on autophagy (Figure 2A). Pathenolide, lactucopicrin, cynaropicrin, dyhydroartemisinin, and molephantin were found to increase the lipidated form of microtubule-associated protein 1 light chain 3 (LC3-II) in glioblastoma cells [22,46,52,72,78], and this process was accompanied by the down-regulation of the autophagic substrate p62 [22,72], extensive cytoplasmic vacuolization [46], and strong rearrangement of the vimentin and α-tubulin cytoskeleton [22]. Notably, despite its stimulatory effect on LC3-II, molephantin blocked autophagic flux and caused autophagosome accumulation in glioblastoma cells due to the defect in autophagosome–lysosome fusion [52]. Consistent with molephantin, xanthatin was also found to inhibit autophagic flux in glioblastoma cells, but its effect on autophagy was strictly negative with the high accumulation of LC-I, inhibition of autophagosome formation, and expression of key autophagic regulators including Beclin-1, ATG5, ATG7, and ATG12 [70], demonstrating a compound-dependent effect of STLs on autophagy. Interestingly, despite the reported anti-autophagic effect, xanthatin induced endoplasmic reticulum (ER) stress in the same cell model, increasing the expression of its key markers, including GRP78, XBP1, CHOP, and ATF4, and enhancing IRE-1α and EIF2α phosphorylation [69]. Given the known association of ER stress with autophagy [114] and its potential as a promising target for glioblastoma therapy [115], the effect of STLs on both autophagy and ER stress requires further detailed studies.

#### 2.3.6. Anti-Glioblastoma Activity of STLs in Combination with Other Drugs and Radiotherapy

The multidirectional effects of STLs against glioblastoma cells described above were mediated by their blocking effect on a number of key signaling pathways, including NF-κB [18,22,55,58,62,66,67,72], STAT3 [15,56,67,83,85], PI3K/Akt/mTOR [22,24,52,65,72,76,77,85], EGFR [60,77,79], and MAPK [22,68] pathways, which play a critical role in glioblastoma pathogenesis [116]. This complex effect determines the possibility of using STLs not only as individual drug candidates but also as components of polychemotherapy regimens for glioblastoma. Indeed, a marked cytotoxic synergy in glioblastoma cells has been demonstrated for combinations of lactucopicrin or dihydroartemisinin with temozolomide [22,78,80] and DMAMCL with cisplatin [65], SHP2 inhibitor SHP099 [24], and X-ray treatment [64]. The results of these studies are discussed below.

The pre-incubation of U87 cells with lactucopicrin at 10 μM for 24 h was found to increase their sensitivity to temozolomide by 23-fold compared to cells treated with temozolomide alone [22]. The Chou–Talalay assay demonstrated that dihydroartemisinin (10–20 μM) also synergistically enhanced the effect of temozolomide in both U87 and U251 cells after 24 h of incubation, and, moreover, increased the sensitivity of temozolomide-resistant glioblastoma cells to its cytotoxic activity [80]. The addition of 0.9 mg/kg dihydroartemisinin administered subcutaneously four days per week to three cycles of temozolomide given orally at 21 mg/kg daily increased the inhibition of LN-Z308 glioblastoma intracranial xenografts in mice by 2.7-fold compared to temozolomide administered alone [78]. The mechanism behind the observed temozolomide-sensitizing effect of dihydroartemisinin included the inhibition of DNA synthesis and phosphorylation of N-Myc downstream-regulated gene 1 protein (NDRG1), which is responsible for temozolomide resistance [78,80].

In the case of DMAMCL, Xi et al. demonstrated its ability to enhance the cytotoxic effect of cisplatin by 3-fold in U118 cells [65]. Using the Bliss independence model, it was found that the maximum synergy was achieved by combining 7.5 μM DMAMCL with 7.5 μM cisplatin, and the effect of this combination was accompanied not only by cytotoxicity but also by pronounced changes in cell morphology and nuclear integrity, reduced migration and invasion of U118 cells, and increased apoptotic rate [65]. Along with cisplatin, DMAMCL showed synergy with SHP099 in patient-derived GSC 1123 and R39 cells, according to the Chou–Talalay method applied by Hou et al. [24]. Consistent with the aforementioned in vitro results, the daily oral administration of DMAMCL at 200 and 100 mg/kg was found to significantly enhance the efficacy of intraperitoneal cisplatin (2.5 mg/kg every three days for a total of six doses) against subcutaneous U118 xenografts [65] and intraperitoneal SHP099 (25 mg/kg daily, excluding weekends) against intracranial 1123 xenografts [24], respectively. The chemosensitizing effect of DMAMCL was associated with its direct interaction with plasminogen activator inhibitor-1 (PAI-1), which is up-regulated in glioblastoma cells and controls their aggressiveness [65], and the inhibition of Akt activation through the down-regulation of AEBP1 [24].

Together with chemosensitization, DMAMCL was found to significantly improve the results of radiotherapy [64]. Li et al. demonstrated that incubation of GL261 cells with DMAMCL for 4 h together with 4 Gy X-ray radiation effectively reduced cell proliferation, lowering the IC_50_ from more than 100 μM in radiation monotherapy to 15.1 μM in the combined group. In addition, 10 μM DMAMCL reduced clonogenic activity in cells exposed to increasing doses of radiation, achieving a radiation enhancement factor of 1.4. The oral administration of DMACL at 50 mg/kg combined with five daily 2 Gy X-ray irradiations significantly attenuated the growth of subcutaneous GL261 xenografts in mice compared to monotherapy. The radiosensitizing effect of DMAMCL is attributed to two key factors, including the enhancement of radiation-induced ROS production with subsequent DNA damage and apoptosis of glioblastoma cells and the stimulation of anti-tumor immunity. The tumor-suppressing activity of DMAMCL is completely abolished in tumor-bearing C57BL/6 mice with homozygous *Rag1* knockout, which prevents the formation of mature B and T cells. In addition, the administration of 75 μg of α-PD-L1 intraperitoneally along with DMAMCL and X-ray radiation not only abrogated tumor growth but also induced long-term immune memory, preventing tumor formation when GL261 cells were reintroduced into mice 30 days after primary tumor regression [64]. These results highlight the promising potential of combined DMACL, radiation, and immunotherapy regimens in the prevention of glioblastoma recurrence.

### 2.4. Permeability of STLs Through Blood–Brain Barrier

The blood–brain barrier (BBB) plays a critical role in maintaining central nervous system (CNS) homeostasis, allowing essential molecules to enter brain tissue and protecting the brain from potentially harmful substances and exogenous xenobiotics [117]. This function of the BBB defines a significant challenge for the treatment of neurological diseases, as effective CNS therapeutics must be able to cross the BBB to reach their intended targets in the brain [118]. Confirming a high anti-glioblastoma potential, a large number of reports have demonstrated the ability of many STLs to effectively cross the BBB using various relevant models. For example, chemoinformatics tools predicted the BBB permeability of ambrosin, artemisinin, and deoxyelephantopin [119,120,121], and in vitro assays, including transwell penetration through HBMEC or MDCK monolayers and the parallel artificial membrane permeation assay (PAMPA), showed a high BBB penetration potential of parthenolide [122] and its dimer [56], costunolide [122], goyazenolide [84], lychnofolide [84], and a 6-*O*,*O*-diacetylbritannilactone [123]. More informative data were obtained in vivo by the LC-MS/MS or positron emission tomography analysis of animal brain tissue after STL injections, demonstrating the successful BBB permeability of DMAPT [47,48], alantolactone [58], DMAMCL [63], and dehydrocostus lactone [18]. In addition, as indirect evidence of brain accumulation of STLs, we can also consider the effective blocking of orthotopic glioblastoma tumor growth by a series of germacranolides [48], eudesmanes [60], guaianolides [15,24,63,66], dihydroartemisinin [80], and enhydrin [86], as well as the high in vivo neuroprotective potential of parthenolide [124] and alantolactone [125]. Thus, the accumulated data indicate the feasibility of using STLs as a platform for the development of novel anti-glioblastoma drug candidates.

### 2.5. Anti-Glioblastoma Efficacy of STLs In Vivo

The pronounced multi-target inhibitory activity of STLs observed in glioblastoma cells in vitro has been extensively verified in glioblastoma mouse models (Table 1). STLs administered intraperitoneally or orally at doses of 5–200 mg/kg were found to effectively block tumor growth in heterotopic glioblastoma xenograft models in mice, demonstrating a high anti-tumor efficacy with a median inhibition rate of 66.5% (Figure 2B). The top five most active STLs included costunolide, DMAMCL, MCL3, alantolactone, and xanthatin, which suppressed tumor growth by 95% [51], 81.5% [63], 79% [67], 70.5% [58], and 69% [69], respectively, compared to control (Figure 2B). Other STLs that inhibited subcutaneous glioblastoma growth were deoxyelephantopin [85], dehydrocostus lactone [18], molephantin [52], dihydroartemisinin [79], and lactones 2 [55] and 4 [83], with inhibition rates ranging from 26.2% to 66.5% (Figure 2B). The pronounced anti-tumor effect of these compounds was accompanied by the absence of organ and brain toxicity, a decrease in cell proliferation [67], and the activation of apoptosis in glioblastoma tissues [51,52,56,69]. In addition, xanthatin and MCL3 increase necrosis area [69] and inhibit angiogenesis [67] in xenograft tumors, respectively, demonstrating the complex effect of STLs on glioblastoma growth in vivo. In addition to the high efficacy in the subcutaneous glioblastoma model, STLs were found to significantly increase the survival of mice with orthotopically implanted glioblastoma, as demonstrated for DMAPT [48], alantolactone [60], DMAMCL [15,24,66], dihydroartemisinin [80], and enhydrin [86] (Table 1). This effect was accompanied by the marked suppression of tumor growth [24,60,63,66,86], a decrease in Ki67 expression [24,66,86], the activation of caspase-3 [80], and a decrease in the infiltration of tumor tissue by pro-tumor M2 macrophages [15]. Taken together, these results clearly confirmed the high anti-glioblastoma potential of STLs.

### 2.6. Clinical Trials of STLs in Oncology Patients

Despite the demonstrated potent inhibitory effect of STLs on glioblastoma growth in vitro and in vivo, to our knowledge, only DMAMCL has successfully completed Phase 1 clinical trials in patients with glioblastoma [126]. In this study, DMAMCL showed satisfactory bioavailability at well-tolerated doses and some effect on tumor progression, resulting in a partial response in 1 patient and disease stabilization in 3 of 12 patients with recurrent glioblastoma evaluated [126]. DMAMCL is currently in Phase 1b/2a and Phase 1 in patients with surgically accessible recurrent glioblastoma multiforme in the USA [127] and China [19], respectively.

Prolonged disease stabilization along with acceptable tolerability and favorable pharmacokinetic profile has been demonstrated for mipsagargin and artesunate in a subset of patients with refractory, advanced, or metastatic solid tumors in Phase 1 studies [43,128]. In addition, artesunate depleted the population of highly proliferative Ki67-positive tumor cells in patients with colorectal cancer (Phase 1) [129] and is now in Phase 2 trials in patients with stage II/III colorectal cancer awaiting surgical treatment [130] and cervical high-grade intraepithelial neoplasia [131]. Furthermore, artesunate was found to regress splenic and lung metastases in combination with dacarbazine in patients with uveal melanoma [132] and demonstrated good tolerability in patients with metastatic and locally advanced breast cancer (Phase 1) [133,134] and anal high-grade squamous intraepithelial lesions (Phase 1) [135]. Notably, three case studies in dogs diagnosed with CD34^+^ spontaneous leukemia, roughly equivalent to human Phase 1 clinical trials, demonstrated the ability of DMAPT to reduce the number of leukemia stem cells in both peripheral blood and bone marrow, independently confirming the ability of STLs to suppress tumor stemness in a non-murine model [136].

Thus, although the aforementioned data demonstrated relatively moderate pharmacological effects of STLs in the early phases of oncology clinical trials, the observed inhibitory properties of these compounds on tumor progression in a subset of patients, favorable pharmacokinetic properties, and the absence of adverse effects confirm the feasibility of further studies with STLs in oncology patients. However, in light of the accumulated data, we believe that STLs should be studied in clinical trials as elements of combined chemotherapy rather than as single agents.

### 2.7. Pharmacological Potential of STLs Against Proneural–Mesenchymal Transition of Glioblastoma Cells

#### 2.7.1. Proneural–Mesenchymal Transition as Promising Target for Glioblastoma Therapy

Proneural–mesenchymal transition (PMT) is a specific type of epithelial–mesenchymal transition (EMT) that occurs in glioblastoma cells and is involved in the regulation of their diffuse growth [7,137]. Given the high association of PMT with glioblastoma severity [3] and the known inhibitory potential of STLs against EMT of various epithelial-type tumor cells [23,138,139,140], a separate chapter of this review is devoted to the effect of STLs on PMT, which has been studied to a much lesser extent. Similar to EMT, PMT is induced in glioblastoma cells by various factors from the tumor microenvironment, including hypoxia [141,142], necrosis [143], therapeutic radiation [144,145], ROS [144], immune cells, including tumor-associated macrophages (TAMs) [145,146] and mesenchymal stem cells (MSCs) [147,148], and cytokines and growth factors [149,150,151]. Both EMT and PMT represent a reversible transformation of cells across a spectrum of phenotypes, rather than a simple dual-state switch, with increasing tumor invasion and metastasis as cells transition toward the mesenchymal state [152,153]. However, because GBM originates from non-epithelial tissues, the extreme points in the PMT spectrum diverge from those in EMT. The transition occurs between the proneural subtype, which expresses DLL3, PDGFRA, OLIG2, and SOX2, and the mesenchymal subtype, which is abundant in CD44, YKL-40, ANXA1, and MET (Figure 3A) [137]. By analyzing the regulatory landscape of PMT, three key elements can be identified, including “hubs”, “regulatory core”, and “stabilizing module”. Two “hubs”, NF-κB and GSK-3β, integrate information from the primary PMT-inducing pathways (Figure 3B). NF-κB is activated in glioblastoma cells by various factors, including mutations in IκBα [154], radiation-induced DNA damage [154], aldehyde dehydrogenase 1A3 (ALDH1A3) [155], the adhesion molecule CD146 [156], and a number of cytokines, such as CXCL1 [151], chemerin [150], TNF-α [157], and TGF-β [146,149]. Once activated, NF-κB stabilizes C/EBPβ in the “regulatory core” [157] and enhances the vasculogenic mimicry of glioblastoma cells by activating the Ephrin-B2/EPHB4 and Notch pathways through the induction of tenascin C (TNC) [158]. The smaller hub, GSK-3β, integrates signals from mutations in NF1 and PTEN [154], as well as from the Wnt [145], sortilin [159], and EGF [160] pathways, playing an important role in glioblastoma pathogenesis. Upon activation, GSK-3β degrades, which stabilizes β-catenin, known as a PMT master regulator [161], and induces WISP1, thereby facilitating PMT through the α6β2-integrin pathway [162,163]. Additional pathways supporting PMT of glioblastoma cells are induced by hypoxia [141,142], necrosis [143], 17β-estradiol [164], CXCL12 [165], C5a [148], bradykinin [147], and the interleukin (IL)-6 family cytokines, including IL-6 [166], leukemia-inhibitory factor (LIF) [167], and oncostatin M (OSM) [168] (Figure 3B).

Transcription factors (TFs) form two “regulatory cores” of PMT, including TAZ/TEAD2 [143,169] and a hierarchical module consisting of C/EBP, STAT3, FOSL2, bHLH-B2, and RUNX1 [170] (Figure 3B). These cores regulate the transcription of proneural and mesenchymal signatures and have nearly non-overlapping sets of regulatory targets [169]. The classical EMT TFs Snail [171], Slug [172], Twist [159], and ZEB1 [148], along with Nrf2 [173] and Fox family TFs FOXD1 [174], FOXM1 [175], and FOXS1 [166], provide additional transregulation of PMT (Figure 3A,B). The “stabilizing module” consisting of deubiquitinases USP3, USP9X, USP10, USP21, and USP36 maintains PMT by preventing the ubiquitin-dependent proteasomal degradation of Snail [171], ALDH1A3 [176], RUNX1 [177], FOXD1 [178], and Slug [179], respectively (Figure 3B).

PMT inhibition shows promise as a strategy against glioblastoma invasion, but it is still in the pre-clinical phase. Compounds from various chemical classes, including lignins [180], nitroimidazoles [181], flavonoids [182,183], ubiquinones [184], benzanilides [185], quinolones [186], halopyridines [187], benzylamines [176], benzodiazines [177], and thiuram disulfides [178], have been shown to inhibit the motility and invasion of glioblastoma cells in vitro and reduce tumor growth in animal xenograft models by targeting key nodes within the PMT regulatory network (Figure 3B). Tetrandrine and DYT-40 target the NF-κB pathway [180,181], while quercetin, nobiletin, and 4-acetylantroquinonol B inhibit signaling pathways linked to GSK-3β, another critical hub in PMT regulation [182,183,184]. HJC0152, nitidine chloride, and YM155 repress STAT3, which plays a central role in the regulatory core of PMT [185,186,187]. WP1130, spautin-1, and disulfiram disrupt the stabilization of ALDH1A3, RUNX1, and FOXD1 by inhibiting the deubiquitinases USP9X, USP10, and USP21, respectively [176,177,178]. All mentioned compounds are considered now as promising anti-glioblastoma drug candidates.

#### 2.7.2. Effect of STLs on Key Regulators of PMT

As shown in Table 1, STLs effectively inhibited the migration and invasion of glioblastoma cells and their spheroidal growth, i.e., processes closely associated with PMT [7]. Indeed, many publications demonstrated a significant inhibitory effect of STLs on the NF-κB signaling pathway, which plays a hub role in PMT (Figure 3B). The blockade of NF-κB nuclear translocation and upstream regulators of this process in glioblastoma cells was demonstrated for alantolactone [58], micheliolide [62], DMAMCL [66], MCL3 [67], dehydrocostus lactone [18], lactucopicrin [22], and cynaropicrin [72]. Notably, IKKβ can be considered as a primary target of STLs: using a pull-down assay, Li et al. demonstrated a direct interaction of DMAMCL with IKKβ [66], and alantolactone and dehydrocostus lactone were shown to form a stable complex with IKKβ in molecular docking simulations [18,58]. The second PMT-related hub, GSK3β, was found to be less susceptible to STLs: it was reported that only DMAMCL and dihydroartemisinin effectively inhibited the expression and phosphorylation of β-catenin, a downstream effector of GSK3β, in glioblastoma cells [65,79].

Consistent with the suppression of PMT-associated hubs, STLs effectively modulated the expression of key PMT-associated markers in glioblastoma cells, including the down-regulation of vimentin by molephantin [52], parthenolide dimer [56], micheliolide [62], DMAMCL [65], lactone 4 [83], and enhydrin [86], the down-regulation of Snail by DMAMCL [65] and N-cadherin by molephantin [52], micheliolide [62], and enhydrin [86], and the up-regulation of E-cadherin by molephantin [52], parthenolide dimer [56], DMAMCL [65], and enhydrin [86]. In addition, alantolactone, micheliolide, MCL3, and dihydroartemisinin effectively inhibited the expression of mesenchymal type-associated matrix metalloproteinases MMP-2 and MMP-9 [58,59,62,67,79], and alantolactone decreased the expression of YAP [60], an important regulator of PMT [3] (Figure 3C). Interestingly, despite a high anti-PMT activity, alantolactone and dihydroartemisinin were found to inhibit the expression of proneural SOX2 and CD133 in glioblastoma cells [60,78], which may indicate the ability of these compounds to affect only specific compartments of the PMT-associated regulome, which requires further investigation.

In addition to inhibiting NF-κB signaling, STLs were found to suppress a number of other signaling axes involved in PMT regulation (Figure 3C), including the following:The blockade of STAT3, as shown for parthenolide dimer [56], MCL3 [67], lactone 4 [83], and DMAMCL (moreover, DMAMCL was found to directly interact with STAT3) [15];PI3K/Akt/mTOR pathway inhibition by molephantin [52], DMAMCL [24,65], lactucopicrin [22], dihydroartemisinin [76,77], and deoxyelephantopin [85];The inhibition of MAPK by brevelin A [68], xanthatin [70], lactucopicrin [22], and cynaropicrin [72];The suppression of EGFR by alantolactone [60] and dihydroartemisinin [77], as well as HIF1α and Smad2/3 by MCL3 [67] and enhydrin [86], respectively.

It should be noted that, despite the demonstrated effects of STLs on key regulators and processes associated with PMT induction (Figure 3C), published reports have only demonstrated the suppressor effect of STLs on the basal proneural–mesenchymal balance in glioblastoma cells, whereas their effect on PMT induced by relevant stimulators, which is more informative for understanding anti-glioblastoma potency [188], has not been investigated to our knowledge. Only Hou et al. demonstrated the ability of DMAMCL to inhibit TGF-β-induced proliferation and spheroid growth of patient-derived glioblastoma stem cells [24], and enhydrin was found to decrease TGF-β expression during PMT blockade in U87 and LN229 cells [86]. Thus, investigating the effects of STLs on PMT induced by various glioblastoma severity-related factors (e.g., TGF-β, EGF, hypoxia, etc.) remains an extremely important research goal for future work.

#### 2.7.3. The Association of Protein Interactome of STLs with PMT

Finally, to estimate the pharmacological potential of STLs with respect to inductor-mediated PMT, the PMT-related gene association network was reconstructed (Figure 4) from two independent datasets, including (i) the proneural and mesenchymal Verhaak gene signatures obtained from patient glioblastoma material characterized by the expression of various glioblastoma severity-associated factors [189] and (ii) the direct protein targets of STLs selected from published data (Table 2) (for detailed information on the methods used, see the Appendix A).

The topological analysis of the obtained interactome clearly confirmed the expediency of further detailed investigation of STLs in stimulated PMT models: more than half of the analyzed direct targets of STLs were found to be associated with a PMT signature with a high Maximal Clique Centrality (MCC) score (Figure 4), an important topological characteristic demonstrating the hub position of nodes within the evaluated interactome [190]. The top five targets of STLs most associated with the PMT gene signature included STAT3, NF-κB, JAK1, JAK2, and EGFR (Figure 4), which is consistent with the data discussed above (Figure 3B,C). The results obtained can be used to design further studies on the anti-PMT activity of STLs. Considering the hub positions of a number of direct targets of STLs, some of the inducer-mediated PMT models can be considered as the most sensitive to STLs and promising for further analysis, including PMT stimulated by EGF (target: EGFR), IL6 (targets: STAT3 and JAKs [191]), IL1 (target: TAK1 [192]), or bacterial toxins (target: NLRP3 inflammasome [193]).

**Table 2 biomedicines-13-00133-t002:** Direct protein targets of STLs.

Type	Compound	Protein	Сonstant, μM	Method	Ref.
Endorepoxide-bearing	Artemisinin	MD2	K_D_ ^1^ = 2.6	Fluorescence titrations, thermal shift assay	[194]
	Artemisitene	FDFT1	K_D_ = 165	Thermal shift assay, SPR ^2^	[195]
	DAMN	ERRα	-	TR-FRET	[80]
		PI3K-β	-	Computational approaches	[196]
Eudesmanes	Alantolactone	AKR1C1	K_D_ = 11.8	SPR, enzyme activity assay	[197]
	Isoalantolactone	STAT3	K_D_ = 100	Thermal shift assay, SPR	[198]
Germacranolides	Costunolide derivative D5	PKM2	K_D_ = 0.018	Thermal shift assay, SPR	[199]
	Costunolide	CaMKII	K_D_ = 21.57	DARTS ^3^, thermal shift assay	[200]
		CDK2	K_D_ = 32.02	DARTS, thermal shift assay, SPR	[201]
		TrxR1	-	SPR, enzyme activity assay	[202]
	Elephantopinolide A Cis-scabertopin Elephantopinolide F	GSTP1	-	Thermal shift assay	[50]
	Eupalinolide B	TAK1	-	Thermal shift assay	[203]
	Parthenolide	USP7	-	Thermal shift assay, SPR, enzyme activity assay	[204]
		FAK1	-	Proteomics, enzyme activity assay	[205]
		HSP72	-	LC-MS/MS	[206]
		NFκB	-	EMSA	[207]
		USP47	IC_50_ ^4^ = 24.97	Thermal shift assay, enzyme activity assay	[208]
	Parthenolide dimer	PKM2	-	Thermal shift assay	[56]
Guianolides and pseudoguaianolides	Arglabin	EGFR	-	Phospho-RTK array	[209]
	Argyinolide S	JAK1	-	DARTS	[210]
	Dehydrocostus lactone	TCTP	K_D_ = 5.33	SPR	[211]
	DMAMCL	PAI-1	K_D_ = 2310	Thermal shift assay, SPR, pull-down	[65]
		STAT3	-	Pull-down	[15]
		IKKβ	-	Pull-down, LC-MS/MS	[66]
	Micheliolide	PKM2	-	Pull-down assay	[21]
	1,6-*O*,*O*-diacetylbritannilactone	NLRP3	-	Thermal shift assay	[212]
	Bigelovin	JAK2	IC_50_ = 44.24	Enzyme activity assay	[213]
	Brevilin A	STAT3	K_D_ = 0.01	SPR	[214]
		IKKα/β	-	Thermal shift assay, LC-MS/MS	[215]
	Britannin	GSK-3β	K_D_ = 30.1	Enzyme activity assay, SPR	[216]
	Helenalin	NF-κB	K_D_ = 4.8	SPR	[217]
Others	4	STAT3	-	Nuclear magnetic resonance	[83]
	Arteannuin B	UBE2D3	K_D_ = 1.2	Thermal shift assay, DARTS, LC-MS/MS	[218]
	Chloranthalactone B	NLRP3	K_D_ = 10.3	Pull-down, DARTS, thermal shift assay	[219]
	Deoxyelephantopin	Hsp90α	-	DARTS	[220]
		PPARγ	-	Enzyme activity assay	[221]
	Isodeoxyelephantopin	TrxR1	-	Enzyme activity assay	[222]
	Tatridin A	PGK1	IC_50_ = 3.76	DARTS, enzyme activity assay	[223]

^1^ K_D_—dissociation constant; ^2^ SPR—surface plasmon resonance; ^3^ DARTS—drug affinity responsive target stability; ^4^ IC_50_—half maximal inhibitory concentration.

## 3. Conclusions and Future Perspectives

STLs are multi-target small compounds of natural origin with proven anti-glioblastoma potential. Scientific data accumulated over the last decade demonstrate the ability of STLs to effectively penetrate the BBB and suppress glioblastoma cell proliferation and viability both in vitro and in vivo through massive mitochondrial imbalance with subsequent oxidative stress and apoptosis induction, the inhibition of cell energy metabolism, the activation of autophagy, and cell cycle arrest. In addition, STLs were found to effectively affect the proneural–mesenchymal balance in glioblastoma cells, contributing to their depletion of mesenchymal features, which contributes to the suppression of tumor cell migration and invasion. The bioinformatic analysis performed by us in the framework of this review suggests the need for further investigation of the anti-PMT activity of STLs using inductor-mediated PMT models to more deeply understand the mechanism of their anti-glioblastoma activity.

## Figures and Tables

**Figure 1 biomedicines-13-00133-f001:**
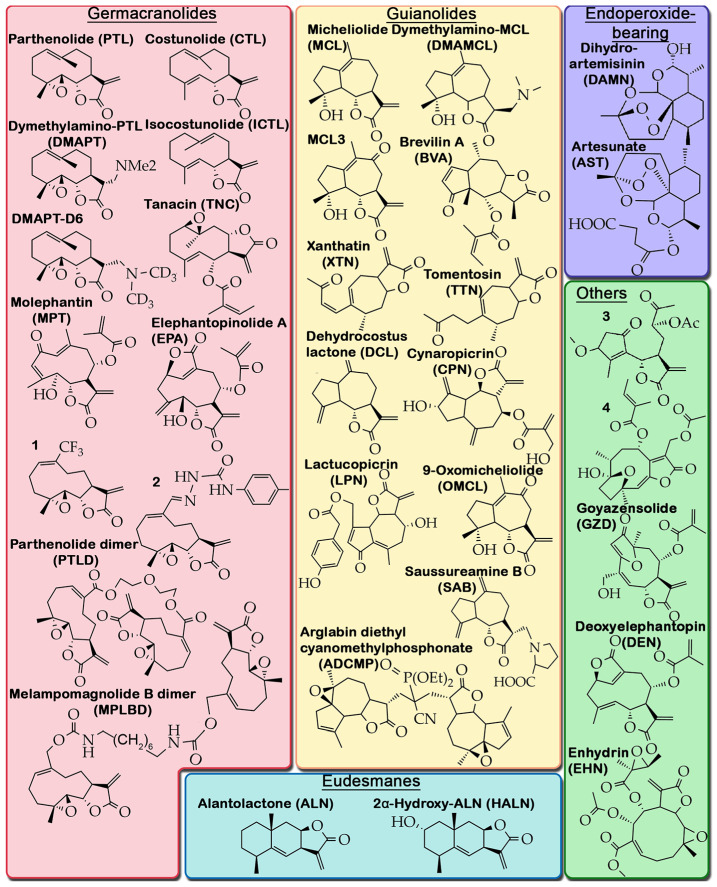
Structures of natural STLs and their derivatives with anti-glioblastoma activities.

**Figure 2 biomedicines-13-00133-f002:**
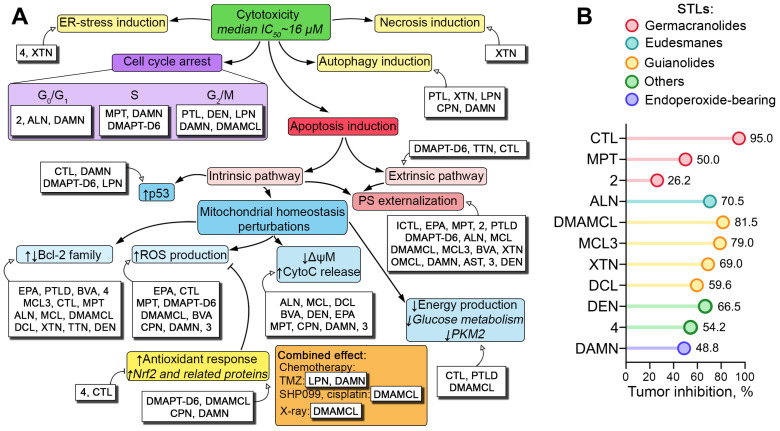
Anti-glioblastoma effects of natural STLs and their derivatives. (**A**) Effect of STLs on key processes associated with glioblastoma cell death. Downward (↓) and upward (↑) bold arrows indicate down-regulation and up-regulation of the processes, respectively. Dull arrows (┴) indicate inhibition. (**B**) In vivo anti-glioblastoma effects of STLs in subcutaneous xenograft models. Different colors represent different types of STLs.

**Figure 3 biomedicines-13-00133-f003:**
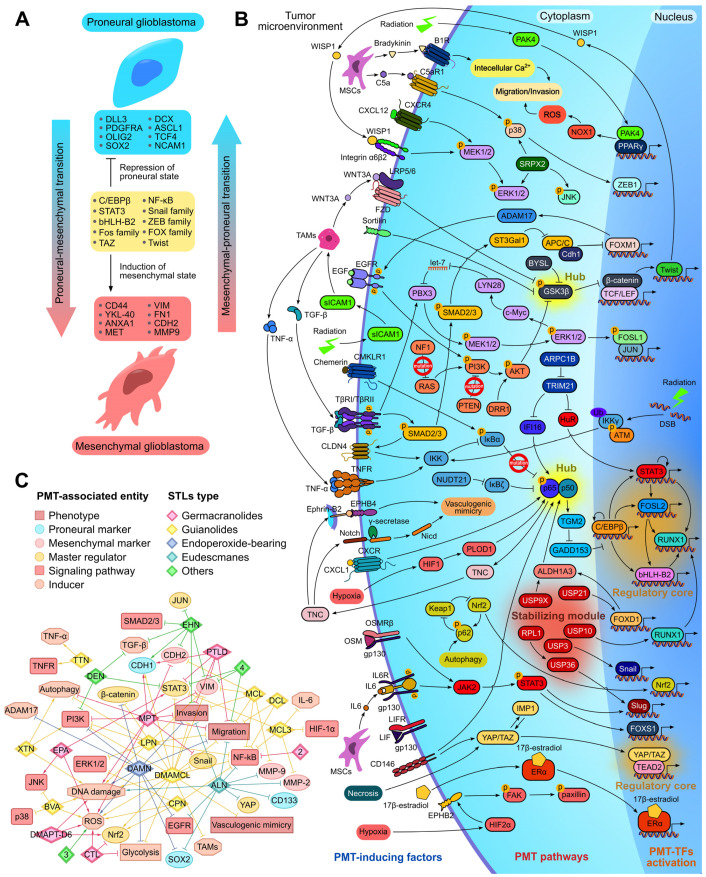
Proneural–mesenchymal transition (PMT) as a target for STLs. (**A**) Cellular plasticity in glioblastoma. In the tumor microenvironment, glioblastoma cells can switch between proneural and mesenchymal states, which differ in morphology and marker expression. The transition to the mesenchymal state, known as PMT, results in an invasive phenotype and is regulated by a subset of transcription factors (TFs), including both classical EMT TFs such as Slug, Snail, and Twis, and PMT-specific TFs such as TAZ, STAT3, and bHLH-H2. (**B**) Pathways that induce PMT. Several factors from the tumor microenvironment, including hypoxia, necrosis, inflammation, and soluble growth factors, can induce PMT. These factors activate a complex regulatory network of signaling pathways in which NF-κB and GSK-3β serve as key interconnected “hubs” (shown in yellow). This network ultimately activates PMT TFs, which are organized in a hierarchy with a “regulatory core” of PMT-specific TFs (highlighted in orange) and supporting TFs. In addition, several deubiquitinases form a “stabilizing module” (marked in red) that maintains PMT by extending the half-life of PMT TFs. (**C**) STL-PMT interaction network. Based on a manual literature review, STLs mediate various PMT-associated cell phenotypes, molecular markers, master regulators, signaling pathways, and inducing factors.

**Figure 4 biomedicines-13-00133-f004:**
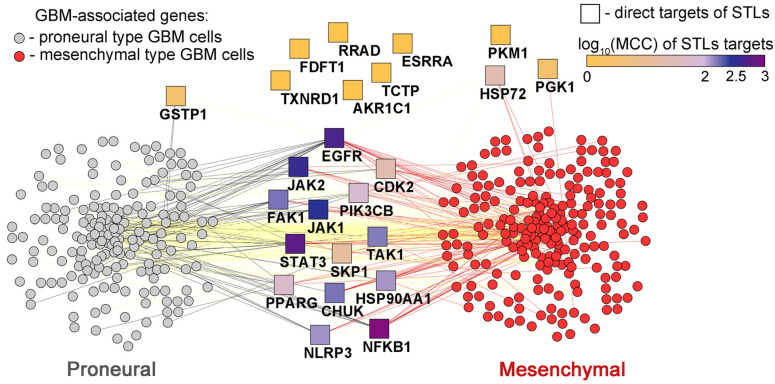
The involvement of direct protein targets of STLs in glioblastoma-related regulome. Proneural and mesenchymal-associated gene signatures were obtained from The Molecular Signatures Database (MSigDB) and are shown as gray and red circle nodes, respectively. Experimentally verified primary protein targets of STLs from published data (shown as square nodes) were added to the glioblastoma-related regulome. MCC—the Maximal Clique Centrality (MCC) score of STL targets within the glioblastoma-related regulome, calculated using the cytoHubba plugin in Cytoscape. Red and black edges indicate interaction of STL targets with mesenchymal and proneural genes, respectively.

## Data Availability

Upon reasonable request, the corresponding author will provide the data generated during this study.

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
