# Peer review of "Sesquiterpene Lactones as Promising Anti-Glioblastoma Drug Candidates Exerting Complex Effects on Glioblastoma Cell Viability and Proneural–Mesenchymal Transition"

_biomedicines, 2025, doi:10.3390/biomedicines13010133_

Round 1

Reviewer 1 Report

Comments and Suggestions for Authors

This manuscript overviews the past decade's research on Sesquiterpene triterpenoids (STLs) as anti-glioblastoma agents, highlighting their potential in targeting this aggressive brain cancer by inducing cell death and inhibiting tumor spread. The authors also provide a detailed discussion regarding to the cellular processes susceptible to STLs' effects. However, the reviewer still has several concerns, which requires a further modification 

1. A more detail description to demonstrate the possible reasons of how STLs trigger the mitochondrial apoptosis pathway through the activation of Caspase-3/-7 is suggested.

2. In my opinion, Figure 2 is not intuitionistic for reading as a scientific manuscript. 

4. Table 1 is too comprehensive for a clear reading. Some parts are incompletely displayed. Please make a double-check and simplify the table.

5. As a review article, the full names of some specialised terms are suggested to demonstrate at their first appearance. e.g., MCL, OPA1, and Mfn1/2.

6. Several previous strategies were succeeded to inhibit glioblastoma cells in preclinical trials, but failed in clinical trials. Please analysis this point more detail regarding to the potential usage of STLs. 

7. Page 21, what is "SLTs"?

Author Response

Dear Reviewer #1,

We are very grateful for your time and valuable comments, which help us to improve the manuscript. Answering a number of your questions has further deepened our understanding of the mechanisms of anti-glioblastoma potential of sesquiterpene lactones and strengthened the feasibility of further investigation of these compounds as glioblastoma-targeting agents. Please allow us to respond to your comments point by point.

  1. A more detail description to demonstrate the possible reasons of how STLs trigger the mitochondrial apoptosis pathway through the activation of Caspase-3/-7 is suggested.

Authors: Corrected. Indeed, a deeper analysis of the text describing the pro-apoptotic effects of STLs revealed a lack of information on possible molecular mechanisms of apoptosis induction. Our additional analysis of published data revealed that the central mechanism of apoptosis triggering under STLs treatment is their ability to stimulate mitochondrial transitory pore opening as a result of modulating the expression of Bcl-2 family proteins and their direct interaction with mitochondria and cytochrome C, followed by activation of the caspase-9 - capsase-3 apoptotic axis. Text with this information has been added to the manuscript (see p. 5, lines 164-180).

  1. In my opinion, Figure 2 is not intuitionistic for reading as a scientific manuscript

Authors: Corrected. Despite the active use of the word cloud plot (Figure 2A) in bioinformatics to illustrate the most enriched terms, we understand that many researchers find this type of data representation to be of little use in scientific manuscripts. In consideration of your valuable comment, we have removed this plot from Figure 2. In addition, in the lollipop plot showing in vivo tumor inhibition of STLs, we have used a better font and provided inhibition values for the convenience of readers (see p. 5, Figure 2B). We hope that this version of Figure 2 is acceptable for the current review.

  1. Table 1 is too comprehensive for a clear reading. Some parts are incompletely displayed. Please make a double-check and simplify the table

Authors: Dear Reviewer #1, we appreciate your comment. In fact, when creating this table, we tried to briefly encode all the valuable information from the analyzed material on the mechanisms of action of STLs and their effects on key processes and proteins associated with glioblastoma pathogenesis, which resulted in the formation of a rather voluminous block. Despite the relative difficulty of reading this table, it can be used by scientists in the field of bioactive STLs as a source of valuable information to search for poorly explored areas (e.g., to identify processes and proteins that are sensitive to some STLs and not explored in other STLs). We are concerned that this functionality will be lost with the simplification of this table. Therefore, we have decided to leave Table 2 unchanged. We hope for your understanding. However, if you insist on changing Table 2, we will do so in the next revision step.

  1. As a review article, the full names of some specialised terms are suggested to demonstrate at their first appearance. e.g., MCL, OPA1, and Mfn1/2

Authors: Corrected. We apologize for this omission. For the convenience of readers, we have added a list of abbreviations into the manuscript (please, see pp. 24-25, lines 661-694). Thank you very much.

  1. Several previous strategies were succeeded to inhibit glioblastoma cells in preclinical trials, but failed in clinical trials. Please analysis this point more detail regarding to the potential usage of STLs

Authors: Corrected. Dear Reviewer #1, we are very grateful for your attention to our article and your valuable suggestion. In fact, the original version of our review contained only a few sentences about clinical trials of STLs in oncology, which is not enough to understand the potential usability of these compounds in the future. Considering this point, we have added section 2.6 entitled "Clinical trials of STLs in oncology patients" (see pp. 17-18, lines 436-466).

  1. Page 21, what is "SLTs"

Authors: Corrected. This was an unfortunate typo (see p. 21, line 612).

We hope that a corrected version of the manuscript will be acceptable for publication in Biomedicines.

Sincerely,

Dr. Andrey Markov

Reviewer 2 Report

Comments and Suggestions for Authors

In this review article entitled ‘Sesquiterpene lactones as promising anti-glioblastoma drug candidates exerting complex effects on glioblastoma cell viability and proneural-mesenchymal transition’, the authors have summarised the literature on the mechanisms of different types of sesquiterpenes as a therapeutic approach for glioblastoma. It is a well-organised review, but lacks some details from a clinical perspective.

The combination of sesquiterpenes with the guideline therapies temozolomide and radiotherapy should be presented in more detail with specific citations.

Possible radiosensitising effects should be mentioned in the text.

The effects of sesquiterpenes on the proliferation of glioblastoma stem cells should be discussed from a mechanistic perspective.

The specificity of these molecules on glioblastoma should be described in detail for toxicity and for effects on other cancers.

Author Response

Dear Reviewer #2,

We are very grateful for your time, careful analysis of our review, important comments, and positive evaluation of our work. Having studied the bioactivity of natural metabolites for the past ten years, our research group would like to contribute to the understanding of the subtle mechanisms of the anti-glioblastoma properties of sesquiterpene lactones (STLs) and hope that this review will be useful to researchers in the field of bioactive nutraceuticals. Let us respond to your comments point by point.

  1. The combination of sesquiterpenes with the guideline therapies temozolomide and radiotherapy should be presented in more detail with specific citations

Authors: Corrected. Thank you for your valuable suggestion. Given the potential of STLs as part of a combination therapy for glioblastoma, a more detailed description of the combined effect of STLs with chemotherapeutic agents and X-ray irradiation has been added to the manuscript according to your comment (see p. 15-16, lines 328-388, section 2.3.5).

  1. Possible radiosensitising effects should be mentioned in the text

Authors: Corrected. Indeed, the original version of our review contained only a very brief description of the radiosensitizing effect of STLs without mentioning possible mechanisms of action and limitations. Following your comment, a paragraph describing these data has been added to the manuscript (see p. 16, lines 370-388).

  1. The effects of sesquiterpenes on the proliferation of glioblastoma stem cells should be discussed from a mechanistic perspective

Authors: Corrected. Given the critical importance of glioblastoma stem cells (GSCs) in glioblastoma recurrence, the description of the modulating effect of STLs on GSCs has been greatly expanded to a separate section 2.3.1 (please see p. 13, lines 191-208). Thank you very much.

  1. The specificity of these molecules on glioblastoma should be described in detail for toxicity and for effects on other cancers

Authors: Dear Reviewer #2, we agree that for a more complete understanding of the subtle mechanisms of antitumor action of STLs, it is necessary to consider their effect not only against glioblastoma cells, but also against tumors of other types. However, given the specificity of our review and its focus on describing the anti-glioblastoma properties of STLs, we decided not to include this information in the text of the article. In Chapter 2.1, we have attempted to provide references to key reviews and experimental articles devoted to the study of the antitumor potential of STLs against non-glioblastoma tumor cells (see p. 2, lines 95-96). We appreciate your understanding. If you insist on adding this information, we will do so in the next stage of revision.

Thank you for your comments!

We hope that a corrected version of the manuscript will be acceptable for publication in Biomedicines.

Sincerely,

Dr. Andrey Markov

Round 2

Reviewer 1 Report

Comments and Suggestions for Authors

Most of my concerns have been revised except Table 1. However, the authors' response is acceptable from my view. Therefore, I suggest to accept the present version.